

# Prediction of active ingredients in *Salvia miltiorrhiza* Bunge. based on soil elements and artificial neural network

Yu Liu[1], Ke Wang[1,2], Zhu-Yun Yan[1], Xiaofeng Shen[3] and Xinjie Yang[4]

[1] State Key Laboratory of Characteristic Chinese Medicine Resources in Southwest China, School of Pharmacy, Chengdu University of Traditional Chinese Medicine, Chengdu, Sichuan, China
[2] School of Big Data and Artificial Intelligence, Chengdu Technological University, Chengdu, Sichuan, China
[3] Institute of Medicinal Plant Development, Chinese Academy of Medical Sciences, Beijing, China
[4] School of Pharmacy, Shaanxi University of Chinese Medicine, Xianyang, Shaanxi, China

## ABSTRACT

The roots of *Salvia miltiorrhiza* Bunge. are commonly used in the treatment of cardiovascular diseases, and tanshinones and salvianolic acids are its main active ingredients. However, the composition and content of active ingredients of *S. miltiorrhiza* planted in different regions of the soil environment are also quite different, which adds new difficulties to the large-scale and standardization of artificial cultivation. Therefore, in this study, we measured the active ingredients in the roots of *S. miltiorrhiza* and the contents of rhizosphere soil elements from 25 production areas in eight provinces in China, and used the data to develop a prediction model based on BP (back propagation) neural network. The results showed that the active ingredients had different degrees of correlation with soil macronutrients and trace elements, the prediction model had the best performance (MSE = 0.0203, 0.0164; $R^2$ = 0.93, 0.94). The artificial neural network model was shown to be a method that can be used to screen the suitable cultivation sites and proper fertilization. It can also be used to optimize the fertilizer application at specific sites. It also suggested that soil testing formula fertilization should be carried out for medicinal plants like *S. miltiorrhiza*, which is grown in multiple origins, rather than promoting the use of "special fertilizer" on a large scale. Therefore, the model is helpful for efficient, rational, and scientific guidance of fertilization management in the cultivation of *S. miltiorrhiza*.

# INTRODUCTION

*Salvia miltiorrhiza* Bunge. is a persistent herb of the genus *Salvia* of the mint family, Lamiaceae (*Chinese Flora Commission of the Chinese Academy of Sciences, 1974*). In China, Japan, and the United States (*Su et al., 2015*), its dried roots are widely used as one of the most commonly used traditional medicinal herbs to treat cardiovascular diseases, especially angina pectoris and myocardial infarction (*Wang et al., 2017*). Researches have shown that diterpenoid quinones and hydrophilic phenolic acids are its principal bioactive components (*Mei et al., 2019*). As cardiovascular disease is a common and frequently-occurring disease

Corresponding author
Zhu-Yun Yan,
yanzhuyun@cdutcm.edu.cn

(*Jagannathan et al., 2019*), the wild resources are unable to meet the ever-increasing market demand. Currently, 19 provinces in China, including Shandong, Henan, Sichuan, Hebei, Hubei, Jiangsu, Shanxi, and Shaanxi (*Lu, 2019*), have been introduced and cultivated *S. miltiorrhiza*, but there are significant differences in the content of active ingredients in the same species planted in different regions (*Huang et al., 2019*; *Yang et al., 2011b*). Soil is the material basis for the survival of plants, and the abundance and deficiency of major and trace elements in soil affect plant growth and development and physiological and biochemical metabolism, which also affect the yield and the composition and content of active ingredients in medicinal plants (*Zhang et al., 2018*). Fertilization is the primary way often used in agricultural production to improve soil nutrition, and proper fertilization is an important measure to ensure the yield and quality of medicinal plants. Therefore, analyzing the relationship between soil elements and active components of medicinal plants and establishing related models can provide data support and technical guidance for selecting suitable sites and fertilization management in the cultivation of *S. miltiorrhiza*.

Although the use of models based on the relationship between soil environment and crop quality or the quality of medicinal plants to predict suitable land, crop yield, and quality has received widespread attention, most studies have used regression analysis models to predict the relationship between the two (*Li et al., 2020*). Regression analysis models are built on the assumption of some idealized linear relationship between predictor and response variables. However, environmental factors show very high intra- and inter-individual variability, which means that the biological responses of plants in response to the environment are uncertain and nonlinear in nature (*Gago et al., 2010*). Therefore, many biological interactions cannot be explained by simple stepwise algorithms or exact formulations, especially in complex or noisy data. Artificial neural network methods have proven effective in solving such problems in different disciplines (*e.g.*, *in vitro* culture (*Hesami & Jones, 2020*), remote sensing studies (*Wang et al., 2019*)). The artificial neural network methods can learn and create nonlinear and complex relationships and can flexibly solve the complex problems of multiple interacting variables (*Bayat et al., 2019*). It is one of the best techniques for extracting information from inaccurate and nonlinear data (*Caselli et al., 2009*). Among them, BP neural network model is the most widely used neural network model in the fields of economics, engineering, and botany (*Armaghani et al., 2018*; *Chen et al., 2019*). The BP neural network has been used to monitor crop growth and crop yield prediction (*Akbar et al., 2018*; *Wang et al., 2019*). However, few reports on the application of the BP neural network predict active ingredients of medicinal plants based on soil elements. Therefore, in this study, we took *S. miltiorrhiza* as the research object, measured the active ingredients in the roots of *S. miltiorrhiza*. and the contents of rhizosphere soil elements from 25 production areas in eight provinces in China, analyzed the relationship between them, and established a BP neural network model for the prediction of active ingredients using soil elements as input values, which expands the application of artificial neural network methods in the field of medicinal botany and also provides references for its application in other directions in the field of medicinal plant cultivation, and verifies the feasibility of using artificial neural network model to effectively improve the accuracy of the prediction of the content of active ingredients of plants.

## MATERIALS & METHODS

### Sample collection and processing

Roots and rhizosphere soil of *S. miltiorrhiza* (cultivation or wild) were collected from 25 producing areas in eight provinces of China from mid-November to early December 2007 (Fig. 1, Table S1). The samples were collected by the "S" parallel sampling and multi-point mixing method, each sample was collected at 20 to 25 points, and the rhizosphere soil was collected by the root shaking method, and finally, the samples for analysis were obtained by the quadratic method, of which 2 kg/sample of rhizosphere soil and 5 kg/sample of medicinal parts were retained. After the samples were collected and quickly transported back to the laboratory, the herbs were processed routinely, and the soil samples were air-dried and prepared.

### Determination of inorganic elements

The available potassium (K), copper (Cu), zinc (Zn), and manganese (Mn) were determined by atomic absorption spectrophotometry after ammonium bicarbonate-diethylenetriaminepentaacetic acid (AB-DTPA) extraction (Lu, 2000); the available nitrogen (N) was determined by alkaline diffusion method (Lin, 2004); available phosphorus (P) was determined by sodium bicarbonate extraction-molybdenum antimony anti-colorimetric method (Zhao et al., 2020).

### Determination of the content of active ingredients

A high-performance liquid chromatographic method was used to determine the contents of water-soluble components and lipid-soluble components (Yang et al., 2010; Yang et al., 2011a). The working parameters for the determination of water-soluble components was as follows: the column was a phenomenex Gemini $C^{18}$ column (250 × 4.6 mm, 5 $\mu$m, Guangzhou Philomen Scientific Instruments Co., Ltd.); the mobile phases were water-acetonitrile-formic acid (90:10:0.4) (phase A) and acetonitrile (phase B), with gradient elution, phase A: 0~40 min, 100%~70%; phase B: 0~40 min, 0%~30%; the detection wavelength was 280 nm; the flow rate was 1 mL/min; the column temperature was 25 °C; the injection volume was 20 $\mu$L. The working parameters for the determination of liposoluble components were as follows: the column was a Welchrom$^{TM}$ $C^{18}$ column (Analytial 4.6 × 250 mm, 5 $\mu$m, Welch Corporation, USA); the mobile phases were methanol (A phase) and water (B phase), with gradient elution, A phase: 0~25 min, 67%~67%; 25~45 min, 67%~90%; B phase: 0–25 min, 33%–33%; 25–45 min, 33%–10%. The detection wavelength was 270 nm; the flow rate was 1 mL/min; the column temperature was 25 °C; the injection volume was 20 $\mu$L.

### BP (back propagation) neural network

Bootstrap is a statistical inference method based on resampling and data simulation. Due to the nonlinear nature of small sample sizes and the difficulty of characterizing the overall distribution, the Bootstrap method can be applied to improve the estimation accuracy of the model (Wang et al., 2018). Therefore, the Bootstrap method has the potential to be widely applied to modeling estimations of small sample sizes. The steps are described as

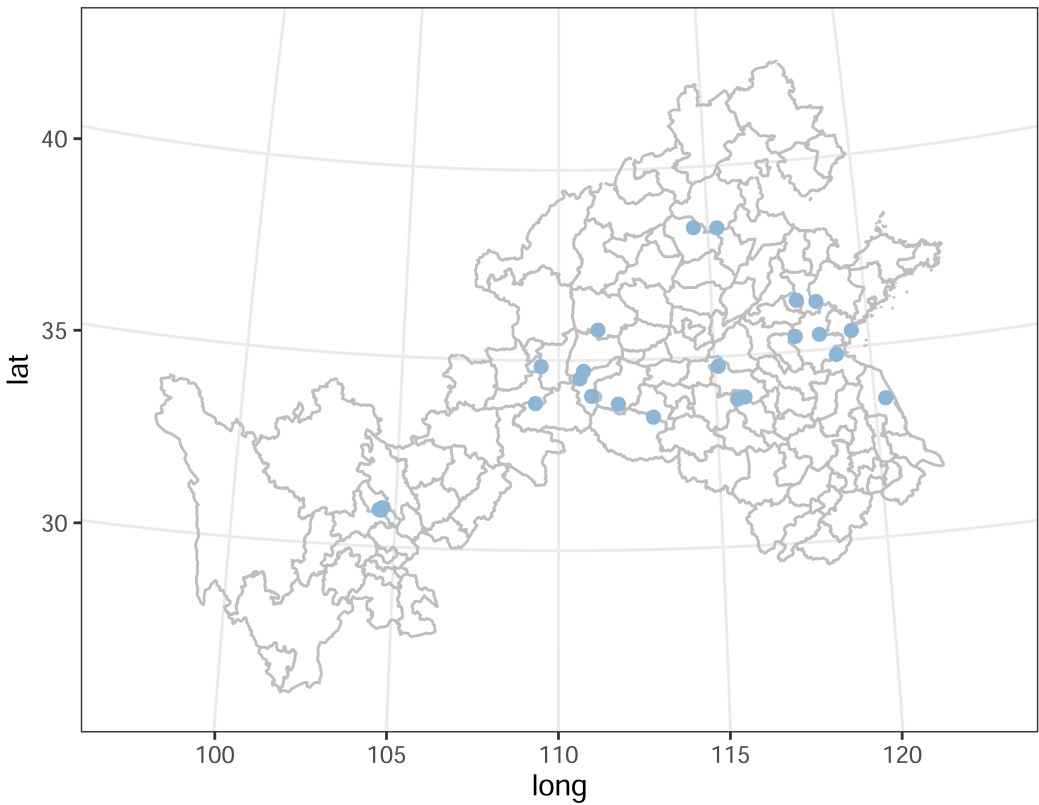

**Figure 1**  Part of China map indicating the location of the sample sites.

follows: (i) perform resampling to select a certain number (given) of samples and to allow repeat sampling; (ii) calculate the given statistics T based on the given samples; and (iii) repeat the above steps N times to gain N number of statistics T (*Wang et al., 2018*). The back propagation neural network proposed by Rumelhart in 1986 is a multi-layer feedforward network trained according to the error back propagation algorithm and is one of the most widely used neural network models (*Rumelhart, Hinton & Williams, 1986*). The BP neural network model algorithm consists of two aspects: the forward propagation of the signal and the back propagation of the error. In other words, the actual output is calculated according to the direction from input to output. However, when the actual output contradicts the expected output, the back propagation of error is performed according to the direction from output to input, and the output error of each layer neuron is calculated layer by layer. Then, the weights and thresholds of each layer are adjusted according to the error gradient descent method to achieve the final output value that can be close to the expected value. The model-building process was described in detail as follows.

In this study, a BP neural network model was constructed using the neural fitting tool (nftool) in MATLAB 2019b mathematical software and trained with soil element parameters as input and the content of plant active ingredients as output. The input variables were as follows: Mn, Cu, Zn, N, P, and K; output variables: water-soluble components (danshensu, protocatechuic aldehyde, caffeic acid, rosmarinic acid, salvianolic acid B, salvianolic acid
A), and lipid-soluble components (dihydrotanshinone I, cryptotanshinone, tanshinone I, and tanshinone IIA). In the solution process, 70% of the samples ($n = 18$) were used to obtain samples using the bootstrap method and build a BP neural network model. The remaining 30% of the samples ($n = 8$) were used to verify the model. First, to prevent the negative impact of different ranges of input variables on the model's efficiency, the input variables in the model were therefore normalized for the specified range (0–1) (Eq. (1)):

$$F_j = \frac{X_j - X_{min}}{X_{max} - X_{min}}.\tag{1}$$

In the equation, $F_j$, $X_j$, $X_{min}$ and $X_{max}$ are the standardized value, original value, minimum value and maximum value of the input variable respectively.

Second, the number of hidden layers and the number of neurons in each hidden layer in a BP neural network impact the overall neural network structure. Currently, many empirical formulas are applied to determine the number of neurons in the hidden layer, and one of these formulas is as follows (Eq. (2)):

$$h = \sqrt{m+n} + a.\tag{2}$$

In the equation, $m$ ($m = 6$) is the number of nodes in the input layer, $n$ ($n = 2$) is the number of nodes in the output layer, and $a$ ($1 \leq a \leq 10$) is a constant.

According to Eq. (2), the number of nodes in the hidden layer was set as an integer between 4 and 12. Eventually, the number of neurons in the hidden layer was determined to be 8 by iterative trials. A neural network model with 6-8-2 structure was finally established (Fig. 2), in which the input layer consisted of 6 neurons corresponding to the 6 input variables, and the output layer had 2 neurons representing the content of active components in the model.

Third, another problem in establishing neural network models is the choice of network learning or training algorithms. Since the Levenberg–Marquardt algorithm minimizes the sum of the error function of the form (Eq. (3)), thus the Levenberg–Marquardt algorithm has the best performance (R-value) compared to other training algorithms (*i.e.,* Bayesian regularization (BR) and scaled conjugate gradient (SCG)) (*Mahmoudi & Mahmoudi, 2014*). Therefore, the Levenberg–Marquardt algorithm will be used to train the network. The training epochs, learning rate, and minimum performance gradient were set as 1000, 0.01, and $1e^{-7}$.

$$E = \frac{1}{2}\sum k(e_k)^2 = \frac{1}{2}\|e\|^2\tag{3}$$

where $e_k$ is the error in the k$^{th}$ exemplar or pattern and $e$ is a vector with element $e_k$.

During the BP training process, a sigmoid function (Eq. (4)) is used to describe the nonlinear relationship between the input and output of each neuron in the hidden layer as follows (*Yi et al. 2007*).

$$f(x) = \frac{1}{1+e^x}.\tag{4}$$

The output $y_j$ of the hidden layer neuron j is calculated by Eq. (5):

$$y_j = \varnothing\left(\sum_{i=1}^{n} W_{ij}X_i + \theta_j\right).\tag{5}$$

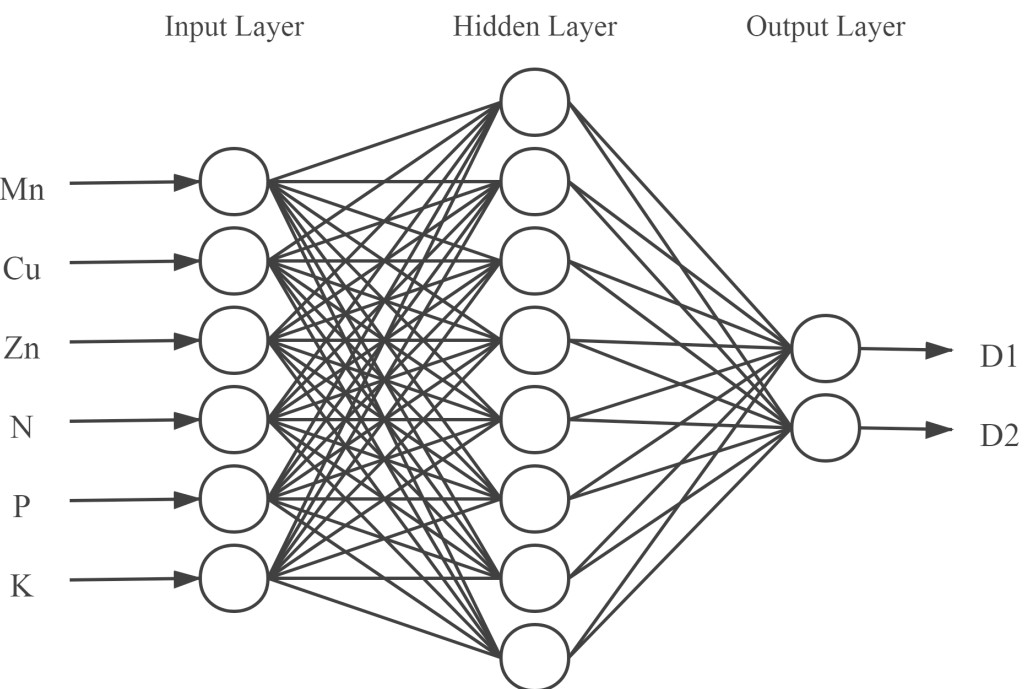

**Figure 2** **Structure of the back propagation neural network.** D1, D2 represents water-soluble components and fat-soluble components.

In the equation, $\phi(x)$ represents the activation function of the hidden layer, $n$ is the number of neurons in the input layer, $X_i$ is the input of the input layer neuron $i$, $W_{ij}$ is the weight from the input layer neuron $i$ to the hidden layer neuron $j$, and $\theta_j$ is the threshold value of the hidden layer neuron $j$.

The output $z_k$ of the output layer neuron $k$ are calculated by Eq. (6):

$$z_k = f\left(\sum_{j=1}^{m} V_{jk}y_j + \alpha_k\right). \tag{6}$$

In the equation, $f(x)$ represents the activation function of the output layer, and $m$ is the number of neurons in the hidden layer, $V_{jk}$ is the weight of the hidden layer neuron $j$ to the output layer neuron $k$, and $\alpha_k$ is the threshold value of the output layer neuron $k$.

The network weights adjustment is defined as follows (Eq. (7)) (*Huang et al., 1996*):

$$\Delta\omega(t) = -\alpha\frac{\partial E}{\partial\omega(t)} + \beta\Delta\omega(t-1) \tag{7}$$

where $\alpha$ and $\beta$ are assumed constants, called the learning rate and momentum factor, respectively, $E$ is the error function (in which MSE was used), $\omega$ is the weight vector, and t is the iteration number (epoch in MATLAB).

Finally, the steps of training can be summarized as follows: (i) applying the input vectors, (ii) calculating the output of the network and comparing it with the corresponding target vectors, (iii) feeding the difference (error) through the network, and (iv) changing the

weights according to the algorithm, which tends to minimize the error. The vectors of the training set are applied sequentially. This process is repeated several times using the entire training set until the error is within acceptable criteria or until the output does not change significantly.

## Predictive performance evaluation

To evaluate the predictive power of the model, the MSE (mean square error) and the $R^2$ (coefficient of determination) were used to evaluate the overall performance of the model (Eqs. (8)–(9)):

$$R^2 = 1 - \frac{\sum_{i=1}^{n}\left(Y_i - Y_j\right)^2}{\sum_{i=1}^{n}\left(Y_i - \bar{Y}_i\right)^2}. \tag{8}$$

$$MSE = \frac{1}{n}\sum_{i=1}^{n}\left(Y_i - \hat{Y}_i\right)^2. \tag{9}$$

In the equation, $Y_i$ is the experimental value of the evaluation model, $\hat{Y}_i$ is the corresponding predicted data, $\bar{Y}_i$ is the average of the experimental data and $n$ is the number of experimental data.

## Data analysis

In this study, correlation plots of the active ingredients of *S. miltiorrhiza* and rhizosphere soil elements were created using the "corrplot" package (version 0.9; https://cran.r-project.org/web/packages/corrplot/index.html) in R. The Pearson's correlation test was used to demonstrate the correlation between the active ingredients of *S. miltiorrhiza* and rhizosphere soil elements. $P < 0.05$ was considered to indicate a statistically significant difference. BP neural network model was created using the neural fitting tool (nftool) in MATLAB 2019b mathematical software (MathWorks Corporation, America). Scatter and regression plots were created using the "ggplot2" package (version 3.3.5; https://cran.r-project.org/web/packages/ggplot2/index.html) in R. The study used the kappa value to evaluate the multicollinearity of characteristic indicators by the "Kappa" package in R. The map plot was created using the "plyr" package (version 1.8.6; https://cran.r-project.org/web/packages/plyr/index.html), "maptools" package (version 1.1-2; https://cran.r-project.org/web/packages/maptools/index.html), and "ggplot2" package (version 3.3.5; https://cran.r-project.org/web/packages/ggplot2/index.html) in R.

# RESULTS

## Correlation analysis of active ingredients and rhizosphere soil elements

In this study, the active ingredients and rhizosphere soil elements of *S. miltiorrhiza* from 25 producing areas in eight provinces of China were determined, and the contents of 6 elements (Mn, Cu, Zn, N, P, K) (Fig. S1) and 10 the kinds of active ingredients (danshensu (D1), protocatechuic aldehyde (D2), caffeic acid (D3), rosmarinic acid (D4), salvianolic acid B (D5), salvianolic acid A (D6), dihydrotanshinone I (D7)), cryptotanshinone (D8),

tanshinone I (D9), tanshinone IIA (D10)) (Fig. S2). From the perspective of linear relationship, Fig. 3 shows the correlation between the content of different active ingredients and soil elements. Most active ingredients are related to major and trace elements (r>0). In addition, the relationships between soil elements (Fig. 3) showed that Mn had a significant positive correlation with Zn ($p = 0.004 < 0.01$); Cu showed a weak correlation with Zn, K ($r < 0.4$); N showed a weak correlation with P ($r < 0.4$); Zn showed a weak correlation with K ($r < 0.4$). It is suggested that there were synergistic properties in the utilization of elements by *S. miltiorrhiza*. Then, the study used the kappa value to evaluate the multicollinearity of typical indicators. A k value below 100 was interpreted as low multicollinearity, and a k value exceeding 1000 indicates high multicollinearity (*Ma et al., 2021*). The kappa value ($k = 480.153$) was exceeded 100. These results indicated multicollinearity among fundamental soil indicators, and linear regression cannot be performed directly. From the perspective of a nonlinear relationship, Figs. 4 and 5 showed the scatter and regression plots between the active ingredients of *S. miltiorrhiza* and the soil elements. The results showed that there was not a simple increase or decrease between the active ingredients of *S. miltiorrhiza* and soil elements, but an inevitable fluctuation would follow, which means that as the content of soil elements increases, the content of active ingredients of *S. miltiorrhiza* will increase and decrease. These results indicated that the active ingredients of *S. miltiorrhiza* had a nonlinear relationship with multiple soil elements, and a particular element index cannot be used to express the effective ingredients of *S. miltiorrhiza*. The artificial neural network methods can learn and create nonlinear and complex relationships and can flexibly solve the complex problems of multiple interacting variables (*Bayat et al., 2019*). It is one of the best techniques for extracting information from inaccurate and nonlinear data (*Caselli et al., 2009*). Therefore, in this study, a BP neural network model with soil elements as input values was established to explore the relationship between soil elements and the active ingredients for predicting the active ingredients.

## Predictive performance of BP neural network model

In this study, a BP neural network model was constructed using the neural fitting tool (nftool) in MATLAB 2019b mathematical software and trained with soil element parameters as input and the content of plant active ingredients as output. The input variables were as follows: Mn, Cu, Zn, N, P, and K; output variables: water-soluble components (danshensu, protocatechuic aldehyde, caffeic acid, rosmarinic acid, salvianolic acid B, salvianolic acid A), and lipid-soluble components (dihydrotanshinone I, cryptotanshinone, tanshinone I, and tanshinone IIA). Through repeated training, a neural network model with 6-8-2 structure was finally established (Fig. 2). In order to evaluate the predictive ability of the established BP neural network model, the MSE (mean square error) and the coefficient of determination ($R^2$) were used to evaluate the overall performance of the model. Ideally. The closer the MSE value is to zero, the closer the $R^2$ value is to 1, which indicates that the average training and testing performance is appropriate. The BP model showed fast training and high simulation accuracy in model testing (Fig. 6A). The relationship between the predicted and measured active ingredients content was favorable, thereby indicating that this model has a basic consistency and a high degree of simulation. A scatter diagram was

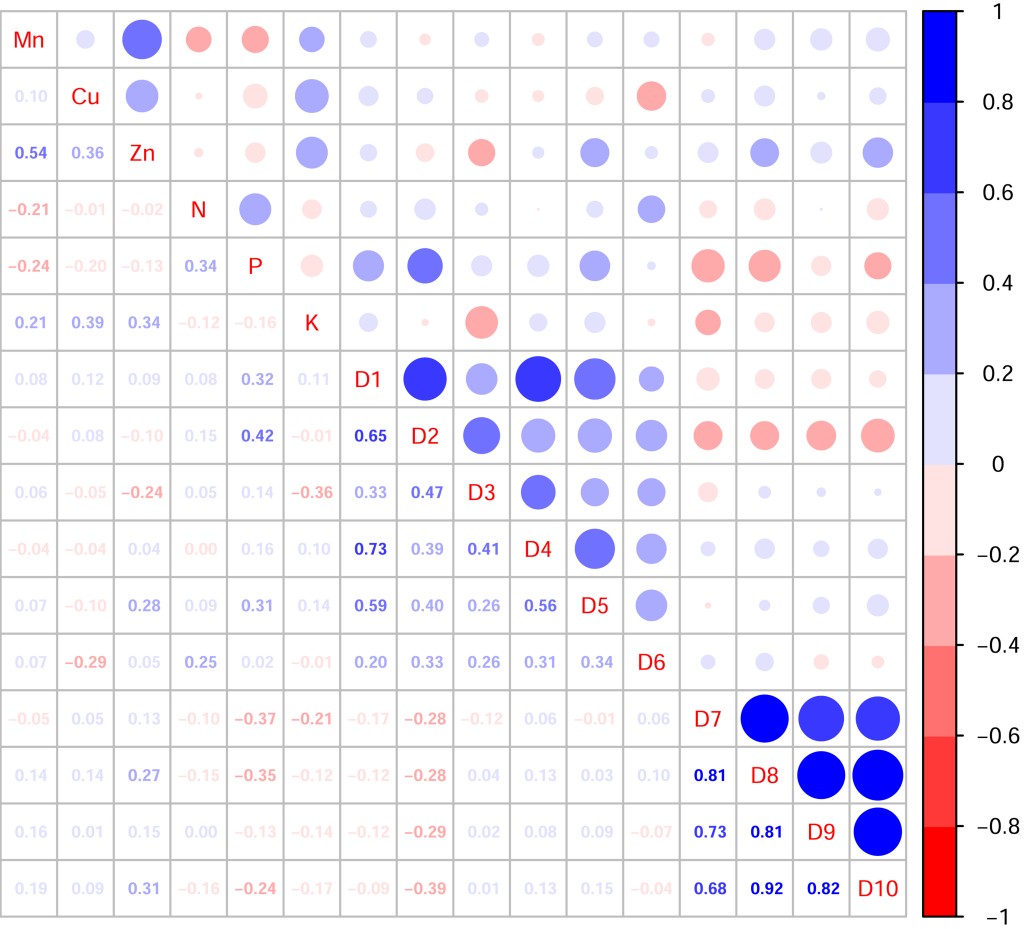

**Figure 3 Correlation diagram between active ingredients and soil elements.** D1-D10 represents danshensu, protocatechuic aldehyde, caffeic acid, rosmarinic acid, dihydrotanshinone I, cryptotanshinone, tanshinone I, tanshinone IIA, salvianolic acid B, salvianolic acid A. Blue indicates positive correlation, red indicates negative correlation. The numbers and the size of the circle represented the correlation coefficients. The larger the area and number, the greater the correlation.

constructed between the predicted and real values of the inversion model. The coefficient of determination R2 between the predicted and real values of the inversion model was 0.93, the linear regression line between the measured and predicted values were close to 1 (*i.e.,* linear), and the MSE (MSE =0.0203) was low. Therefore, the predictive ability of the model was relatively high and showed strong nonlinear fitting ability, indicating that the soil elemental content could be used for accurate inversion of the active ingredient content.

From the results obtained, the empirical equation based on the Levenberg–Marquardt algorithm for predicting active ingredient content in normalized form is Eq. (10):

$$(D1, D2) = \sum_{j=1}^{3} \left[ purelin \left\{ \sum_{i=1}^{8} \sum_{j=1}^{6} logsig \left[ \left( Mnj_1 + Cuj_2 + Znj_3 + Nj_4 + Pj_5 + KJ_6 \right) + b_i \right] \right\} \times Lw_{i,j} + b_{k_i} \right]. \tag{10}$$

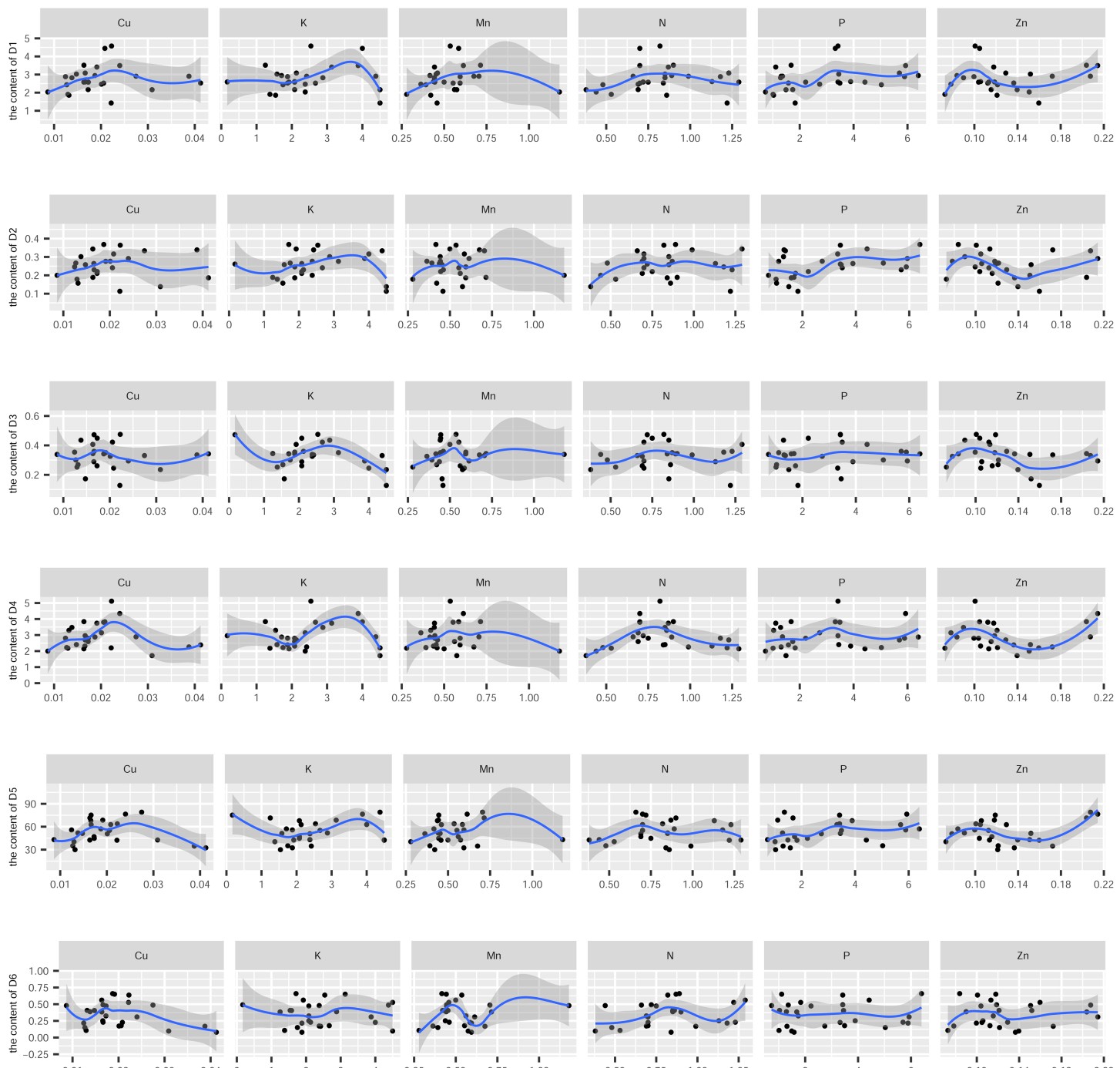

**Figure 4** **The scatter and regression plots between the water-soluble components of *S. miltiorrhiza* and the soil elements.** D1-D6 represents danshensu, protocatechuic aldehyde, caffeic acid, rosmarinic acid, salvianolic acid B, salvianolic acid A.

Peerj

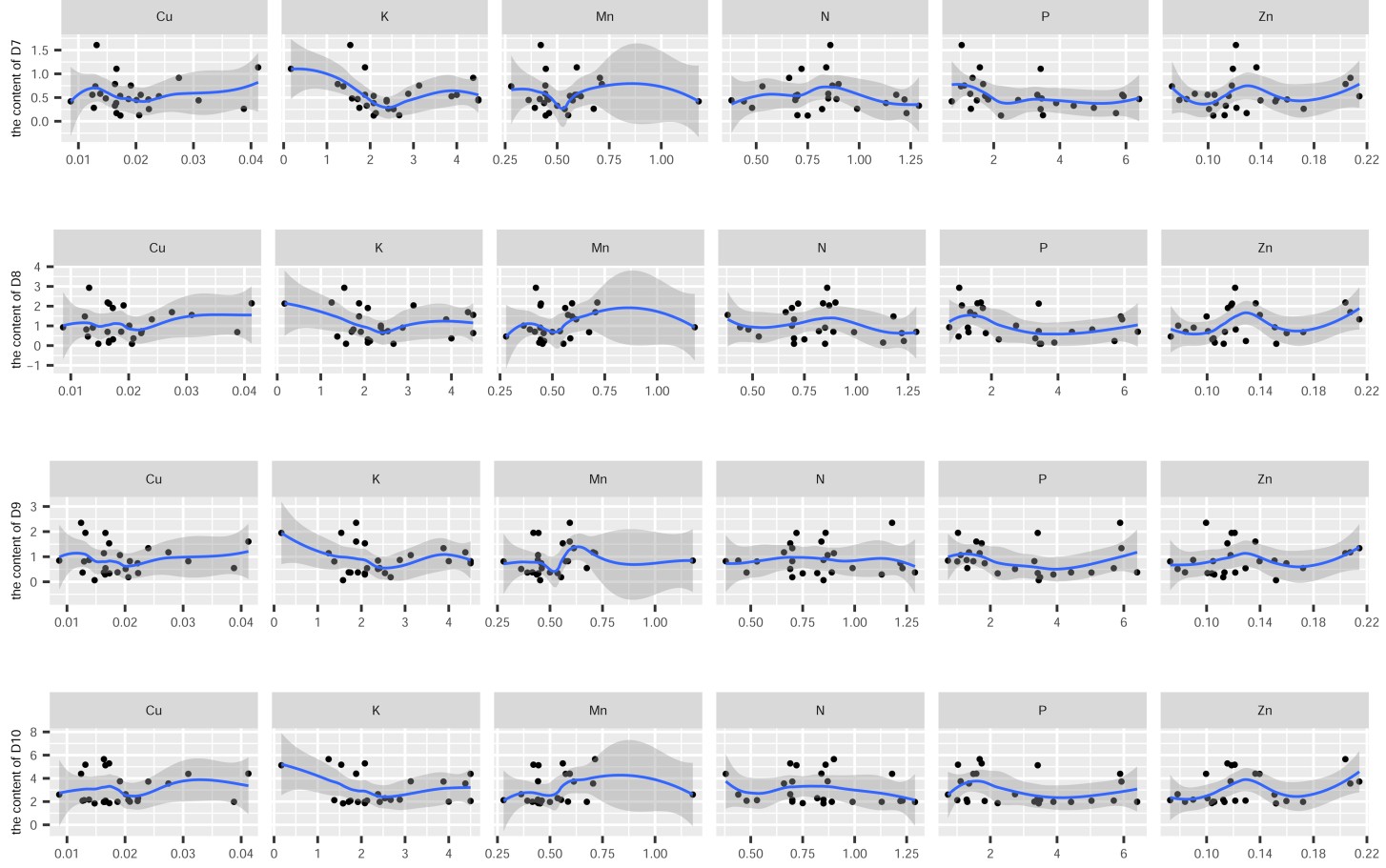

**Figure 5** **The scatter and regression plots between the lipid-soluble components of _S. miltiorrhiza_ and the soil elements.** D7–D10 represents di-hydrotanshinone I, cryptotanshinone, tanshinone I, tanshinone IIA.

The transfer function "*purelin*" correlated the linear relationship between the input and output variables, while "*logsig*" calculated the layer's output from the network input. The variables $j_1, j_2, j_3, j_4, j_5,$ *and* $j_6$ were input weights from the input layer to the hidden layer. Also, $Lw_1, Lw_2$ were hidden layer weights from the hidden layer to the output layer. The variables $b_i$ and $b_k$ were biases connected to the hidden layer neurons and output layer neurons, respectively. The values for these weights and biases for the model were in Table 1.

To further evaluate the model's generalizability, the developed neural network model was tested using 41 new datasets. The new datasets were derived from the results of Zhang's study (*Zhang et al., 2018b*). The correlation coefficient, mean square error, was used to evaluate the generalization ability of the model. A scatter diagram (Fig. 6B) was constructed between the predicted and real values of the inversion model. The coefficient of determination $R^2$ between the predicted and real values of the inversion model was 0.94, the linear regression line between the measured and predicted values were close to 1:1 (*i.e.,* linear), and the MSE (MSE =0.0164) was low. Therefore, the predictive ability of the model was relatively high. Our results indicate that the BP neural network model based

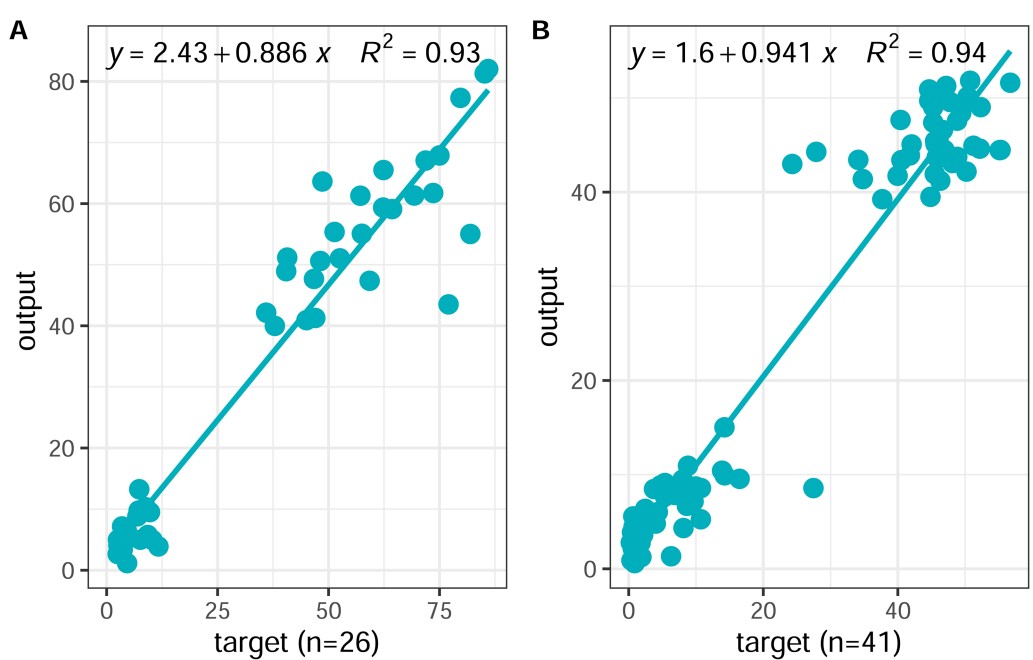

**Figure 6** Scatter plots of the BP neural network outputs *versus* targets values. (A) Scatter plot of outputs *versus* targets values of dataset ($n = 26$); (B) scatter plot of outputs *versus* targets values of new dataset ($n = 41$).

**Table 1** Weights and biases of the model.

| $i$ | Input weights | | | | | | Output layer biases | Hidden layer weights | | Output layer biases |
|---|---|---|---|---|---|---|---|---|---|---|
| | $j_1$(Mn) | $j_2$(Cu) | $j_3$(Zn) | $j_4$(N) | $j_5$(P) | $j_6$(K) | $b_1$ | $Lw_1$ | $Lw_2$ | $b_k$ |
| 1 | 0.996143 | −0.02048 | 2.317871 | −0.37965 | −0.56815 | 0.54681 | −2.080196453 | 1.472587 | 0.699387 | −0.212174053 |
| 2 | 0.838481 | 0.545057 | 1.347549 | 1.166583 | 0.741616 | 0.673065 | −1.217495956 | −0.60922 | 0.526448 | 0.540334675 |
| 3 | −0.88541 | 1.155593 | −0.07306 | −0.36007 | 0.91443 | 0.29891 | −0.874658689 | −0.28682 | 1.10661 | |
| 4 | −0.78139 | 1.617126 | −0.09689 | 1.246001 | 0.545242 | 0.535554 | 1.953430407 | 0.985855 | −1.55521 | |
| 5 | 1.154236 | 1.305964 | −0.34628 | 1.030944 | −1.48709 | −1.23566 | −0.164765862 | −0.18309 | −0.66367 | |
| 6 | −1.56626 | 3.327984 | 1.205375 | 0.906212 | −2.31703 | 1.897674 | 0.667870142 | −0.55526 | 0.804189 | |
| 7 | 0.275443 | 1.076501 | −1.06431 | 1.218909 | 0.808466 | −0.84648 | 1.880952801 | −0.10458 | 1.113067 | |
| 8 | 2.092393 | 0.073101 | −0.1403 | 0.38235 | −0.49332 | 0.539668 | 2.641857041 | 0.501492 | 0.781672 | |

on the content of soil elemental can be a powerful tool for predicting the content of active ingredients of *S. miltiorrhiza*.

## DISCUSSION

Soil is the material basis for the survival of plants. The large and trace elements in the soil can provide nutrients for plants, provide good growth and metabolic status, and enhance their resistance to adversity (*Alam & Naik, 2009*; *Zhang et al., 2018*). These are the basis for the production of secondary metabolites of the plant. Isopentenyl diphosphate (IPP) and its

isomer dimethylallyl diphosphate (DMAPP) are the precursors of all terpenoids (*Ma et al., 2015*). The synthesis of both requires P. The phenylpropanoid pathway and the tyrosine-derived pathway are the main pathways for the biosynthesis of salvianolic acids (*Ma et al., 2015*). The synthesis of phenylalanine, the starting point of the phenylpropane pathway, requires the participation of N. K not only regulates plant water metabolism but also acts as an activator of enzymes involved in respiration and photosynthesis (*e.g.*, glutathione synthase) (*Liu, 2009*). Trace elements (Mn, Cu, Zn) are involved in the biosynthesis of plant active ingredients as cofactors, constituent elements, and activators of some enzymes (*Guo et al., 2005*). Therefore, the biosynthesis of the active ingredients of *S. miltiorrhiza* cannot be achieved without the participation of soil elements. In this study, the results of correlation analysis between soil elements and active ingredients based on the harvesting period of the herb showed that the active ingredients of *S. miltiorrhiza* interacted with several soil elements and showed a nonlinear relationship, which may be different from the results of fertilization experiments, for example, *Han & Liang (2005)* reported that phosphorus fertilization showed a positive effect on the accumulation of danshensu and tanshinone IIA, but this effect was not observed for the effective phosphorus content in this study. Combined with the Mao's research results (*Mao et al., 2009*), it is believed that it is a matter of elemental availability or antagonism in the utilization between elements. Moreover, there were different levels of correlation between the affective states of the elements, suggesting a synergistic or antagonistic effect on elemental uptake by *S. miltiorrhiza*. The results of our study further suggest that soil testing formula fertilization should be implemented for medicinal plants like *S. miltiorrhiza*, which is grown in multiple origins, rather than promoting the use of "special fertilizer" on a large scale. The relationship between fertilizer requirements of medicinal plants and soil supply should be coordinated. The required nutrients should be supplemented in a targeted manner to achieve a balanced supply of nutrients and guide the fertilization management of medicinal plants in an efficient, rational, and rational scientific manner. Therefore, it is imperative to establish a method that can be used for screening suitable cultivation sites and rational fertilization of medicinal plants.

Here, we presented and validated the application of artificial neural networks in predicting the active ingredients of medicinal plants. Artificial neural networks have received more attention in various applications due to their sensitivity, accuracy, non-destructive nature, and rapidity (*Wu, Liu & Lu, 2012*). Many researchers have used artificial neural networks to monitor crop growth and crop yield prediction (*Akbar et al., 2018*; *Wang et al., 2019*; *Yang et al., 2018*), but there are few reports on the use of artificial neural networks to predict active ingredients of medicinal plants based on soil elements. Our results showed that artificial neural network models provided more appropriate predictions of the content of *S. miltiorrhiza* using the site's soil elemental characteristics conditions. We recorded minimal differences between the predicted and observed data (MSE $=0.0203$, $0.0164$; $R^2 = 0.93, 0.94$), and the excellent agreement between the results support the high efficiency of artificial neural networks while demonstrating that the use of mathematical models (*e.g.*, artificial neural networks) to predict the active ingredient content in medicinal plants can reduce the time and cost required for analytical methods.

Artificial neural network models offer significant advantages over traditional mathematical models by using nonlinear network connections and allowing analysis that explores the efficacy of all input variables simultaneously, thereby improving the accuracy of results (*Alam & Naik, 2009*; *Tušek et al., 2018a*; *Tušek et al., 2018b*). However, the capability of artificial neural network models is usually also limited by drawbacks such as slow learning speed, overfitting, and local minima, which suggests that building some hybrid neural network models to reduce their drawbacks and improve their performance is an essential direction for future research.

Many medicinal plants have greater medicinal productivity in their original habitat than in cultivated areas. Soil nutrient characteristics similar to the original habitat must be best suited for the production of active metabolites (*Alam & Naik, 2009*). Based on the results of this study, an artificial neural network model can be used to screen suitable cultivation sites for medicinal plants and simulate soil conditions similar to the original soil conditions through soil management, thus increasing the yield of active ingredients. The specific measures are as follows: firstly, measuring the key nutrient factors of the soil, such as massive elements and trace elements; and finally balancing fertilization, such as targeted supplementation of required nutrients during plant growth and harvest, to achieve a balanced supply of nutrients and to efficiently, rationally and scientifically guide the fertilization management of medicinal plants. However, the soil matrix is a complex organic ecosystem, while the plants' secondary metabolites themselves are a complex physiological process, so the effect of soil on plant active ingredients is far more complex than what is involved in the current study. So future work could also extend this modeling process by using a more inclusive range of biotic and abiotic variables to obtain more accurate estimates.

## CONCLUSIONS

In this study, an artificial neural network model for predicting the active ingredients of *S. miltiorrhiza* using rhizosphere soil elements as input values were developed to localize the effects of these factors on the active ingredients. This expands the application of artificial neural network methods in medicinal botany and provides a reference for other directions in the field of medicinal plant cultivation. The results show that, in combination with soil data, we can use artificial neural network models to successfully predict the content of *S. miltiorrhiza*. This further validates the feasibility that artificial neural network models can effectively improve the accuracy of the prediction of the active ingredient content of medicinal plants. In addition, the model can provide broader applicability for ranch managers, manufacturers, and producers of medicinal plants to screen suitable sites for medicinal plant cultivation in a robust and reproducible manner. It can also optimize the fertilizer application at specific sites and guide the fertilization management of medicinal plants in an efficient, rational, and rational scientific manner.

### Funding

This study was supported by the Natural Science Foundation of China (81973416). The funders had no role in study design, data collection and analysis, decision to publish, or preparation of the manuscript.

### Grant Disclosures

The following grant information was disclosed by the authors:
Natural Science Foundation of China: 81973416.

### Competing Interests

The authors declare there are no competing interests.

### Author Contributions

- Yu Liu conceived and designed the experiments, performed the experiments, analyzed the data, prepared figures and/or tables, authored or reviewed drafts of the paper, and approved the final draft.
- Ke Wang conceived and designed the experiments, authored or reviewed drafts of the paper, and approved the final draft.
- Zhu-Yun Yan conceived and designed the experiments, authored or reviewed drafts of the paper, provide experimental venue, and approved the final draft.
- Xiaofeng Shen and Xinjie Yang performed the experiments, authored or reviewed drafts of the paper, and approved the final draft.

### Data Availability

   The raw measurements are available in the Supplementary File.

### Supplemental Information

Supplemental information for this article can be found online at http://dx.doi.org/10.7717/peerj.12726#supplemental-information.

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
