# Peer review of "Prediction of active ingredients in Salvia miltiorrhiza Bunge. based on soil elements and artificial neural network"

_PeerJ, doi:10.7717/peerj.12726_

## Round 0.1 · original submission · Major Revisions

Noble Authors,

Four experts thoroughly assessed your work. Everyone agreed that the paper has many flaws, and the biggest objection is too low a number of experimental cases which are needed to teach such a network. It is possible to further proceed with this article, but only if all corrections will be introduced.

Reviewer 3 has requested that you cite specific references. You may add them if you believe they are especially relevant. However, I do not expect you to include these citations, and if you do not include them, this will not influence my decision

With best regards,

·

Basic reporting

no comment

Experimental design

no comment

Validity of the findings

no comment

Additional comments

1. The results were mainly obtained by the method of the artificial neural network, thus the description of the data should be more conservative throughout the manuscript.
2. The quality of figure 1 should be improved.
3. A conclusive or abstract graph will be helpful for the general readers.

Reviewer 2 ·

Basic reporting

This paper is written in proper English, suitable for international audience. Paper background and the overall problem are described in clear and comprehensible way. Previous research referred by the authors are relevant and referenced correctly and in consistent form. Paper structure match the PeerJ standards. Figures and tables are adequate, properly described and supplement well the description of obtained results. Authors supplied raw data as well as additional figures. Nonetheless, minor drawbacks appear in several lines - detailed comments are given below.

Lines 45-49.
Please check the completeness of the first part of the sentence ‘Currently, 19 provinces in China, including Shandong, Henan, Sichuan, Hebei, Hubei, Jiangsu, Shanxi, and Shaanxi (Lu 2019), have been introduced and cultivated, but there are significant differences in the content of active ingredients in the same species of S. miltiorrhiza planted in different regions (Huang et al. 2019; Yang et al. 2011b).’ - the object is probably in wrong position.

Please consider such solution: ‘Currently, 19 provinces in China, including Shandong, Henan, Sichuan, Hebei, Hubei, Jiangsu, Shanxi, and Shaanxi (Lu 2019), have been introduced and cultivated S. miltiorrhiza, but there are significant differences in the content of active ingredients in the same species planted in different regions (Huang et al. 2019; Yang et al. 2011b).’

Lines 129-132.
I suggest to revise an order of the sentence ‘Therefore, in the BP training, select the input Xi of the input layer neuron i, the weight Wij from the input layer neuron i to the hidden layer neuron j, the threshold value θj of the hidden layer neuron j, and the output yj of the hidden layer neuron j is calculated by Equation 1:’ since it’s a little bit incomprehensible. The better solution would be to start the sentence with ‘The output yj of the hidden layer neuron j is calculated by Equation 1:’ or something similar, then give the formula followed by description of its all components (a description similar to given in lines 144-145 should be suitable).

Lines 135-137.
The same comment as above.

Lines 212-213.
There is probably lack of object in sentence ‘The water-soluble components (D1-D4, D11, and D12) showed a higher degree with Mn, Cu, Zn, N, P, and K.’ Degree of what? Please review this sentence.

Experimental design

Line 140.
Please check if the formula given in line 140 is correct. Usually the error function is the sum of squares (or sum of squared error known as SSE), which is minimized during training process. The formula given by Authors is rather square of sum, which could give the same results but only if in parenthesis the minuend is higher than the subtrahend. Moreover, Wang et al. (2019) after Rumelhart et al. (1986) claims that ‘The algorithm is a multilayer supervised learning network trained using the methods of mean square error (MSE) and gradient descent, which involves adopting a learning rule with the steepest gradient descent method to achieve the minimum sum of squared error.’, which suggests that formula given by Authors may be just written incorrectly. However, if formula is correct, please include some reference to justify this way of error computing.

Please provide information about transfer (activation) functions between input and hidden layer as well as between hidden and output layer of ANN. What were the functions? identity, sigmoid (S-shape), hiperbolic tangent, sinusoid, negatively exponential, Gaussian? Please provide the formulas or references, if possible. Additionally, information about number of training epochs, learning rate and convergence criterion should be given.

I suggest to explain clearly somewhere in the sub-division BP (back propagation) neural network what authors mean using term ‘Test data’ (it appears in Table 1). Generally, in ANN-related issues 3 kinds of data sets are reported: training set (other words input data or input vectors) - data presented constantly to the ANN in order to realise training process according to appropriate algorithm, validation set (the subset of input data) - data used for supervision of ANN training process, most of all to avoid network overfitting, which is very common in case of BP learning algorithm, and test set - data never used for training but necessary for final evaluation of ANN and calculation of fit-measures. I suppose that term 'Test data' concerns the latter, but very often term ‘validation set’ is used interchangeably with the term ‘test set’ and refers to assessment the performance of a fully-trained model, which might be confusing, especially for readers unfamiliar with ANN terminology. To avoid confusion please define clearly the role of test set or refer to an appropriate source (e.g. Ripley, 1996).

Ripley B.D. 1996. Pattern Recognition and Neural Networks. Cambridge University Press, UK., pp. 354.

Validity of the findings

Lines 146-147 and 223-264
I have a serious objection to the size of training data set, especially taking into account ANN architecture. It is not about the percentage (this is ok) but the quantity of training vectors (18 cases). Figure 6 presents ANN structure with 8 inputs, 16 hidden nodes and 7 outputs which give 8×16×7 = 896 network connections (other words parameters (weights) to select in each iteration). Although selection of weights is realised on the training set, the final task of ANN is reproduction, which means that the real goal of network training is to minimize so-called generalization error, not training error. Generalization error is a function of data density and network complexity measure, called VCdim (Vapnik-Chervonenkis) measure (Vapnik, 1992). A small number of training samples (low data density), with a fixed measure of VCdim, leads to model overfitting and bad generalization, as the approximation task is reduced to interpolation and the model becomes susceptible to reproduction of disturbances as a basic features. According to Hush and Horne (1993) good generalization of ANN may be obtained if number of training vectors exceeds VCdim by at least 10 times. Since there is no simple relationship between VCdim measure and network architecture, it is only possible to estimate its lower and upper limit, which approximately correspond to number of weights connecting input with hidden layer and twice the number of all network weights, respectively. In practice, VCdim measure is assumed as the number of all network weights (number of all network connections). Having ANN architecture proposed by the Authors, minimum 8960 training vectors are required to ensure good generalization abilities of such model. Dataset contains only 26 observations, one for each specific location and plant type (cultivated or wild). Since there are no biological replications for each location, no variability of studied natural phenomenon was captured which also provides to uncertainty of other results presented in this paper. I strongly suggest to include additional training vectors (much more than 18) by making biological replications. With additional data Authors may use also some analytical methods based on known algorithms, e.g. bootstrap, to increase the size of training set – nonetheless these methods need data dispersion (variance). Then I suggest to (a) revise the ANN architecture towards reducing the number of hidden neurons or (b) try different solution, e.g. the set of networks (one for each dependent variable to reduce network connections) or (c) try different kind of network which realizes multidimensional regression and is much more resistant to a limited number of training vectors, e.g. Support Vector Machine for regression (Drucker et al., 1997).

Vapnik V.N. 1992. Principle of risk minimization for learning theory. In: Moody J., Hanson S., Lippmann R., (eds). Advances in Neural Information Processing Systems 4 (NIPS 1991), Morgan Kaufmann, San Mateo, pp. 831 – 838.

Hush D., Horne B. 1993. Progress in Supervised Neural Networks. IEEE Signal Processing Magazine, 10: 8 – 39.

Drucker H., Burges C.C, Kaufman L., Smola A.J., Vapnik V.N. 1997. Support Vector Regression Machines. In: Advances in Neural Information Processing Systems 9 (NIPS 1996), MIT Press, pp. 155 –161.

Line 193.
Since cluster analysis is not a statistical analysis, no statistical tests (like multiple comparisons) are performed to assess the significance of differences between data vectors. In addition there is impossible to perform such comparisons using statistical approach, since there is no data replication in dataset, hence no dispersion measure may be calculated. Therefore, it may be written that Shandong and other producing areas differ in terms of analysed features, but nothing confirms that these differences are significant (this term is reserved for specific, statistical uses). Moreover, please justify the need for conducting cluster analysis. Did its results have any impact on further work on the neural model?

Lines 197-198.
It is about a part of the sentence which starts in line 194. Authors write: ‘…while other active ingredients were correlated with major and trace elements, but not significantly’ - if correlation is not significant (null hypothesis about r = 0 should not be rejected), it means that there is no correlation (linear one of course), thus further consideration of such dependency is not justified.

Line 202-204.
Collinearity is not the same as correlation. Authors did not check collinearity statistics, e.g. VIF, condition index or proportion of variation. Where conclusion about multicollinearity came from? Finally collinearity is not an error and does not exclude possibility of building a valid regression model.

Figures 4 - 5 and corresponding description in lines 209-212.
This comment concern mainly description of B, Fe and less Cu trends in figures 4 and 5. Scatter plots between these soil elements and components of S. miltiorrhiza shows that some points may be ‘suspected’ of being outliers. It is also possible that dataset consists of more than one subpopulation but it is impossible to find out if it is so since dataset contains too few observations. It is risky to discuss a trend when it is formed by only one or two observations, without checking their influence statistics. For the future, please check influence statistics and identify potential outliers or consider data which form different clusters separately to avoid so called Simpson paradox (Simpson, 1951).

Simpson E. H. 1951. The Interpretation of Interaction in Contingency Tables. Journal of the Royal Statistical Society, Series B, 13: 238-241.


Conclusion

In my opinion research issue described here is very interesting, which is one of the strengths of the paper. Another strength is clear, concise and generally reliable description of research tasks (with minor failings).

Nonetheless, this work has serious methodological drawback and that is why I suggest a major revision of this paper. I'm not convinced to the results obtained by the Authors. This is due to the lack of proper representation of analysed phenomenon. At the same time, I strongly emphasize that I have no objection to the method of obtaining data from the collected soil and plant samples, but to the lack of fundamental feature of experimental data - replication, which provide the knowledge about variability/diversity of natural phenomena. Consequently, the authors obtained high value of correlation coefficients which turned out to be insignificant (this is an indication that something is wrong with the data). In addition, it is not possible to check the basic properties of variables (distributions, collinearity, the influence of observations) and to use classical statistical methods to learn them more deeply (the lack of repetition excludes most analyses based on variance). Unfortunately, it is highly likely that the results obtained for the neural network are the effect of model overfitting, what may be seen in Figures 7-10 (too much adjustment to some data and the inability to reproduce the general relationship).

Additional comments

I would like to encouraged the Authors to improve the dataset, revise ANN architecture as I recommended above and repeat procedure of ANN model creating. Please treat this experimental work as a preliminary study of your problem. I will be glad to read the paper again, after improvements.

Reviewer 3 ·

Basic reporting

No comment.

Experimental design

This study is required because the roots of Salvia miltiorrhiza Bge. are used in the treatment of cardiovascular diseases. The article will not be accepted in this format. Some major revisions needed.
• The author described that content of active ingredients of S. miltiorrhiza planted in different regions of the soil environment are different. In my opinion the content of active ingredients not only varies due to some soil factors but also due to environmental factors, altitude, pH of the soil etc. The developed model is based upon soil factors only. Without the environmental factors it is impossible to select the suitable cultivations sites. So if possible develop the model with both soil and environmental factors.
• Give voucher no for each collected areas.
• Add the article: Abdul Akbar, Ananya Kuanar, Raj K Joshi, I S Sandeep, Sujata Mohanty, Pradeep K Naik, Antaryami Mishra, Sanghamitra Nayak (2016) Development of Prediction Model and Experimental Validation in Predicting the Curcumin Content of Turmeric (Curcuma longa L.). Frontiers in Plant Science 7: 1507.

Validity of the findings

no comment

Additional comments

no comment

Reviewer 4 ·

Basic reporting

The experiment was inappropriately designed for the stated goals and research topic.
The authors did not define research hypotheses, but briefly mentioned the purpose in the introduction. The introduction is a loose discussion of the problems that were described in the paper. The data obtained from the experiment do not allow the use of most of the used statistical methods and artificial neural networks.

Experimental design

In designed experiment 26 data records were obtained from 25 locations.
ANN assume learning based on a very large number of responsive data records.
A network model with 8 neurons in input layer, 16 neurons in hidden layer and 7 in output layer was presented. The assumption is correct, but the implementation is not. The number of connections between the input layer and the hidden layer is 128. a minimum of 1280 experimental cases is needed to teach such a network. The authors trained it with 18 samples and verified with 8. This is against the assumptions of the use of ANN.
Correlation values ​​between elements and active ingredients were rightly given in the results. Unfortunately, the obtained data do not meet the assumptions of a normal distribution, which the authors ignore in the study. Figures 4 and 5 show that only a few of the studied variables had a distribution that did not differ from the normal distribution for which the correlation method can be used.

Validity of the findings

On the basis of such a small experiment, it is not possible to generalize the relationships and dependencies between the elements in the tested soil and the content of active substances in the plant.
The authors do not present the source data, but it can be concluded from the description that the only method used correctly was hierarchical cluster analysis. Unfortunately, the methodology does not describe it in detail.

---

## Round 0.2 · Minor Revisions

Dear Authors,

Your corrections to the first version have been approved by all reviewers. One of them, however, asks for some minor adjustments. I kindly ask You to make these corrections.

With best regards,

·

Basic reporting

no comment

Experimental design

no comment

Validity of the findings

no comment

Additional comments

The manuscript has been revised accordingly.

Reviewer 2 ·

Basic reporting

No comment

Experimental design

No comment

Validity of the findings

No comment

Additional comments

In general I'm satisfied with all improvements made by the Authors and I recomend this paper for publication in PeerJ.
My only comment concerns the caption of Figure 6. Please check the captions (A) and (B) for proper singular and plural forms of words 'plots' and 'datasets'. I suppose that each scatterplot represents outputs varsus targets of one dataset where n = 26 or n=41 means the number of observations. Thus, shouldn't these captions be:
'(A) Scatter plot of outputs versus targets values of dataset (n=26); (B) Scatter plot of outputs versus targets values of new dataset (n=41).' ?

Reviewer 3 ·

Basic reporting

no comment

Experimental design

no comment

Validity of the findings

no comment

---

## Round 0.3 · accepted · Accept

Noble Authors,

Thank you very much for making the corrections suggested by the reviewers. The work in its current version may be published in PeerJ.
With best regards,